# *Chenopodium murale* Juice Shows Anti-Fungal Efficacy in Experimental Oral Candidiasis in Immunosuppressed Rats in Relation to Its Chemical Profile

**DOI:** 10.3390/molecules28114304

**Published:** 2023-05-24

**Authors:** Samah A. El-Newary, Asmaa S. Abd Elkarim, Nayera A. M. Abdelwahed, Elsayed A. Omer, Abdelbaset M. Elgamal, Wael M. ELsayed

**Affiliations:** 1Medicinal and Aromatic Plants Research Department, Pharmaceutical and Drug Industries Research Institute, National Research Centre, 33 El Bohouth St., Dokki-Giza-Egypt, Giza 12622, Egypt; 2Chemistry of Tanning Materials and Leather Technology Department, National Research Centre, Giza 12622, Egypt; 3Chemistry of Natural and Microbial Products Department, Pharmaceutical Industries Institute, National Research Centre, Giza 12622, Egypt; 4Department of Chemistry of Microbial and Natural Products, Pharmaceutical and Drug Industries Research Division, National Research Centre, Giza 12622, Egypt; algamalgene@yahoo.com; 5Chemistry of Medicinal Plants Department, Pharmaceutical and Drug Industries Research Division, National Research Centre, Giza 12622, Egypt

**Keywords:** *Chenopodium murale* juice (CMJ), anti-fungal efficacy, immunosuppressed rats, LC-MS/MS analysis, IL-17, INF-γ

## Abstract

*Chenopodium murale* (Syn. *Chenopodiastrum murale*) (*amaranthaceae*) is used in the rural Egypt to treat oral ulcers in newborn children. The current study aimed to discover new natural products suitable for treating candidiasis disease with minimal side effects. Characterization of bioactive compounds by LC-QTOF-HR-MS/MS from *Chenopodium murale* fresh leaves’ juice (CMJ) was carried out in order to elucidate their potential anti-fungal and immunomodulatory effects in oral candidiasis in immunosuppressed rats. An oral ulcer candidiasis model was created in three stages: (i) immunosuppression by drinking dexamethasone (0.5 mg/L) for two weeks; (ii) *Candida albicans* infection (3.00 × 10^6^ viable cell/mL) for one week; and (iii) treatment with CMJ (0.5 and 1.0 g/kg orally) or nystatin (1,000,000 U/L orally) for one week. Two doses of CMJ exhibited antifungal effects, for example, through a significant reduction in CFU/Petri (236.67 ± 37.86 and 4.33 ± 0.58 CFU/Petri), compared to the *Candida* control (5.86 × 10^4^ ± 1.21 CFU/Petri), *p* ≤ 0.001. In addition, CMJ significantly induced neutrophil production (32.92% ± 1.29 and 35.68% ± 1.77) compared to the *Candida* control level of 26.50% ± 2.44. An immunomodulatory effect of CMJ at two doses appeared, with a considerable elevation in INF-γ (103.88 and 115.91%), IL-2 (143.50, 182.33%), and IL-17 (83.97 and 141.95% Pg/mL) compared with the *Candida* group. LC-MS/MS analysis operated in negative mode was used for tentative identification of secondary (SM) metabolites based on their retention times and fragment ions. A total of 42 phytoconstituents were tentatively identified. Finally, CMJ exhibited a potent antifungal effect. CMJ fought *Candida* through four strategies: (i) promotion of classical phagocytosis of neutrophils; (ii) activation of T cells that activate IFN-γ, IL-2, and IL-17; (iii) increasing the production of cytotoxic NO and H_2_O_2_ that can kill *Candida*; and (iv) activation of SOD, which converts superoxide to antimicrobial materials. These activities could be due to its active constituents, which are documented as anti-fungal, or due to its richness in flavonoids, especially the active compounds of kaempferol glycosides and aglycone, which have been documented as antifungal. After repetition on another type of small experimental animal, their offspring, and an experimental large animal, this study may lead to clinical trials.

## 1. Introduction

An oral ulcer is an ulcer that occurs on the mucosal membrane of the oral cavity. An ulcer is a tissue defect that has penetrated the epithelial–connective tissue border, with its base at a deep level in the submucosa or even within muscle or periosteum [1]. Some common factors may aggravate mouth ulcers; these include smoking, foods high in acidity or spice, biting the tongue or inside of the cheek, braces, poorly fitting dentures and other apparatus that may rub against the mouth and gums, defective fillings, stress or anxiety, hormonal changes during pregnancy, puberty, menopause, medications including beta-blockers and pain killers, and genetic factors. In addition, several conditions, such as hypo-salivation, diabetes mellitus, poor oral hygiene, and prolonged antibiotic and corticoid therapy, can predispose people to oral candidiasis. Some people may develop ulcers due to a different medical condition or a nutritional deficiency. Conditions such as celiac or Crohn’s disease, vitamin B12 or iron deficiency, or a weakened immune system may trigger ulcers to form [2].

Epidemiological surveys indicate that *Candida* organisms are present as commensals in the oral cavities of approximately 40% of healthy subjects. *Candida albicans* is explicitly carried as a commensal organism in the mouths of about one third of the population. Infection in the mouth is characterized by white discolorations on the tongue, and around the mouth and throat. Irritation may also occur, causing discomfort when swallowing. Thrush appears common in infants. It is not considered abnormal in infants unless it lasts longer than a few weeks [3].

Consequently, *C. albicans* is a major cause of oral and esophageal infections in immunocompromised patients. It affects up to 90% of patients with human immunodeficiency virus infection or AIDS. The expression of *C. albicans’* virulence in the oral cavity strongly correlates with impairment of the immune system [4]. Many anti-fungal medications are available for the treatment of *candidal* infection. These drugs are polyenes and azoles. However, polyene drugs (amphotericin B and nystatin) may be associated with the incidence of hepatotoxicity and nephrotoxicity. Meanwhile, azole drugs (itraconazole and fluconazole) may be linked to *C. albicans* resistance. Therefore, a search for new and effective products to treat this fungal infection is needed to avoid these drawbacks [5]; new and effective drugs are required to treat this fungal infection.

Natural and synthetic glucocorticoids remain at the forefront of anti-inflammatory and immunosuppressive therapies. They are commonly indicated for treating acute and chronic inflammations. However, long-term use of oral glucocorticoids is associated with severe side effects, including osteoporosis, metabolic disease, and increased risk of cardiovascular disease. Many immunosuppressive actions of glucocorticoids have been discovered. Glucocorticoids decrease immune response through several mechanisms, including (i) suppression of T cells; (ii) inhibition of B cells’ proliferation and antibody production; (iii) suppression of neutrophils’ functions of adhesion, chemotaxis, phagocytosis, and the release of toxic substances; (vi) suppression of macrophages, which become less able to phagocytose opsonized things; and (v) reduction of prostaglandin and leukotriene production [6].

CMJ is a plant species in the *amaranthaceae* known by the common names nettle-leaved Goosefoot, Australian spinach, salt green, and sowbane. This plant is widespread worldwide, particularly in tropical and subtropical areas. It is a common weed in fields and at roadsides [7]. The plant is an annual herb, reaching 70 cm in height with an erect stem. The plant is usually red or red-streaked green and covered with mealy white hairs; its stem is ribbed with a few side branches, and the plant is leafy, with green foliage. Its leaves alternate between oval and triangular leaves; they are toothed and broad, smooth on the upper surface and powdery on the undersides. The inflorescences are granular clusters of spherical buds. The plant’s flowers are inconspicuous, with no petals. The fruit has a single-seeded utricle; the seeds are black nutlets, keeled and pitted [8]. The plant is edible and nutritious. In the UAE, the leaves are used as a salad green. The plant has been used as (i) a diuretic; (ii) a mild purgative; (iii) an emollient; (iv) an anthelmintic; (v) a tranquilizer; (vi) a carminative; (vii) an aphrodisiac; and (viii) a tonic. Additionally, it has been used to treat digestive disorders, including peptic ulcers, dyspepsia, flatulence, and hemorrhoids. It has also been used to treat seminal weakness, cardiac disease, ophthalmopathy, hepatopathy, spleenopathy, pharyngopathy, and general debility. Additionally, *Chenopodium* is used to treat anxiety, depression, hair loss, and cough [9]. A few studies have been published on the biological activities of *C. murale,* exploring its anti-inflammatory and analgesic activities [10], anti-fungal effect [11], antibacterial effect [12], hypotensive action [13], and hepatoprotective activity [14]. *C. murale* contains flavonoids, saponins, and terpenoids [15]. Phytochemical evaluation of *Chenopodium murale*, Linn. revealed flavonoids, essential oils, sterols, steroidal estrogen-like substances, alkaloids, and coumarins [9,16].

Fresh CMJ is used in rural Egypt to treat oral ulcers in newborn children. Therefore, this study was carried out to study the anti-candidal activity of CMJ and its relation to the immune system using an oral candidiasis model. We also studied the chemical profile of CMJ using HPLC/QTOF-HR-MS/MS analysis.

## 2. Results and Discussion

### 2.1. Anti-Candidal Effect of CMJ

#### 2.1.1. Anti-Microbial Effect of CMJ

Before initiating the study, oral cavity cultures of each rat were performed, and no *C. albicans* organisms were found. Just before the treatment, all groups of infected animals were sampled. The oral swabs were all positive for the presence of *C. albicans*, with a mean of CFU/swab of 5.85 × 10^4^ ± 0.77 (Table 1). After CMJ treatment at 0.5 g/kg for seven days, the mean CFU/swab decreased significantly. Meanwhile, CMJ at 1.0 g/kg for seven days led to a more significant reduction in mean CFU/swab. However, there was a similar significant decrease in mean CFU/swab in the nystatin group and in the treatment group with the high concentration of CMJ. Consequently, the degree of reduction in CFU versus control was 99% for CMJ at 1.0 g/kg for nystatin (when used at the recommended dose of 1 mL/kg/day, which equals 100,000 units).

#### 2.1.2. Immunomodulatory Effect of CMJ

Compared to the negative control, hemoglobin and red blood cell count were not affected either by *Candida* infection or CMJ administration. However, dexamethasone-induced immunosuppressed rats experienced significant thrombocytopenia. The platelet count of the ulcer control group significantly decreased by about 55.38% compared to the negative control (*p* ≤ 0.001). On the contrary, CMJ administration exhibited significant thrombocytosis, while the platelet count of CMJ (0.5 g/kg)-treated and CMJ (1.0 g/kg)-treated groups was significantly elevated to 234.99 × 10^9^ ± 5.06 and 321.98 × 10^9^ ± 8.33 platelet/L, compared to the ulcer control group (194.75 × 10^9^ ± 5.41 platelet/L). The same trend was noticed in the positive groups, in which platelet counts significantly increased compared with the negative control (Table 2).

*Candida* infection caused a condition of inflammation, in which ulcer group animals produced a large number of leukocyte cells (13.30 × 19^9^ ± 1.50 cell/L); this number was larger than the negative group by about 54.65% (*p* ≤ 0.001). Additionally, the leukocyte differential of the ulcer group recorded remarkable alternation, where the lymphocyte% significantly reduced (59.07% ± 4.22). Meanwhile, the monocyte% was significantly induced (9.56% ± 0.79), with no significant change in neutrophil, eosinophil, and basophil percentages compared with the corresponding values in the negative group. CMJ force-feeding significantly depleted the leukocyte count and remarkably modulated its differentiation, where neutrophil% increased at the expense of the lymphocyte, monocyte, and basophil%. The leucocyte counts of the CMJ (0.5 g/kg)-treated and CMJ (1.0 g/kg)-treated groups were significantly depleted compared with that of the ulcer group, by about 66.17 and 64.14%, respectively. Additionally, more than in the ulcer group, the neutrophil percentages were significantly elevated by about 24.23 and 34.65%. Meanwhile, the last two groups’ lymphocyte, monocyte, eosinophil, and basophil% were remarkably reduced compared with those of the ulcer group (*p* ≤ 0.001). CMJ administration decreased leukocyte count and increased neutrophil% more effectively than nystatin, compared to the ulcer group. The leucocyte count of the positive groups was also considerably reduced. The leucocyte differential was remarkably ameliorated toward the level of the neutrophil%, compared to the negative group. CMJ (0.5 g/kg)-treated and CMJ (1.0 g/kg)-treated groups appeared with a leucocyte (6.97 × 10^9^ ± 0.18 and 7.23 × 10^9^ ± 0.44 cells/L) and a lymphocyte% (59.17% ± 3.67 and 54.94% ± 0.81) which were lesser than the negative group, and a neutrophil% (30.73% ± 2.99 and 35.07% ± 1.01) and monocyte% (6.99% ± 0.78 and 6.88% ± 0.51) larger than those of the negative group. There were no significant differences in the eosinophil and basophil percentages between the positive and negative groups.

In Figure 1A, the oral ulcer control group appeared with a low INF-γ concentration (140.90 ± 2.25 Pg/mL), which is about 32.62% less than that of the negative group (209.11 ± 2.32 Pg/mL), *p* ≤ 0.001. CMJ caused significant immune enhancement, where the INF-γ concentration was significantly elevated in the CMJ (0.5 g/kg)-treated and CMJ (1.0 g/kg)-treated groups, at a level of 287.27 ± 4.19 and 304.22 ± 3.76 Pg/mL, respectively, compared to the ulcer group (140.90 ± 2.25 Pg/mL). The INF-γ concentration of the two groups was higher than that of the nystatin group. The same trend was recorded in the INF-γ concentration of CMJ (0.5 g/kg) and CMJ (0.5 g/kg)-positive groups, being significantly increased to 279.18 ± 4.91 and 291.74 ± 2.95 Pg/mL, respectively, compared to the negative control (209.11 ± 2.32 Pg/mL). Compared to the negative control, the INF-γ concentration of all groups that were administered CMJ was higher.

The data in Figure 1B illustrate that *Candida* infection significantly reduced IL-2 from 72.34 ± 1.08 Pg/mL in the negative control to 45.56 ± 0.87 Pg/mL in the ulcer control (*p* ≤ 0.001). Compared to the ulcer control, the IL-2 of CMJ (0.5 g/kg)-treated and CMJ (1.0 g/kg)-treated groups was significantly elevated by about 143.50 and 182.33%, respectively. CMJ at 0.5 and 1.0 g/kg significantly increased IL-2, more than nystatin (110.94 ± 1.18, 128.63 ± 1.02, 94.49 ± 0.75 Pg/mL, respectively). In addition, CMJ significantly raised IL-2 in the positive groups, where the IL-2 of CMJ (0.5 g/kg)-positive and CMJ (1.0 g/kg)-positive groups increased more than the IL-2 of the negative group (96.64 ± 1.35, 114.86 ± 1.16, and 72.34 ± 1.08 Pg/mL). The IL-2 of all groups administered CMJ was higher than the IL-2 of the negative control.

The data in Figure 1C show that the IL-17 of the ulcer group was remarkably less than that of the negative group, by about 38.06% (*p* ≤ 0.001). On the contrary, CMJ significantly increased the IL-17 of the CMJ (0.5 g/kg)-treated and CMJ (1.0 g/kg)-treated groups (210.19 ± 2.53 and 276.42 ± 2.97 Pg/mL with 83.97 and 141.94% increase) compared to the ulcer control (114.25 ± 2.88 Pg/mL). CMJ at 0.5 g/kg boosted the IL-17 of candidiasis rats more than nystatin (276.42 ± 2.97 and 228.76 ± 2.29 Pg/mL). Additionally, the IL-17 of CMJ (0.5 g/kg)-positive and CMJ (1.0 g/kg)-positive groups was significantly elevated to 205.45 ± 3.22 and 214.94 ± 3.37 Pg/mL, showing an 11.38 and 16.53% increase from the negative group (184.46 ± 2.93 Pg/mL).

### 2.2. Antioxidant Impact of CMJ

The results in Table 3 show that oral ulcer induction caused significant depletion in the CAT activity of the serum and oral mucosa, but not the spleen, compared to the negative control (*p* ≤ 0.001).

CMJ showed an antioxidant effect, and significantly increased CAT activity. The CMJ (0.5 g/kg)-treated and CMJ (1.0 g/kg)-treated groups recorded CAT activity higher than that of the ulcer group by about 146.25 and 172.12% for the serum, 123.73 and 135.85% for oral mucosa, 24.09 and 63.81% for the spleen, respectively. Nystatin significantly elevated the CAT activity of the serum, oral mucosa, and spleen, compared to the ulcer group, but this elevation was less than that of CMJ. Additionally, CMJ significantly increased the CAT activity of positive groups, where the CAT activity of the serum, oral mucosa, and spleen in CMJ (0.5 g/kg-positive and CMJ (1.0 g/kg)-positive groups was higher than that of the negative control (*p* ≤ 0.001). CMJ augmented CAT activity in all groups the administered it, more so than the negative control.

The data in Table 3 demonstrate that *Candida* infection significantly reduced the SOD activity of the serum (226.52 ± 3.94 U/mL), oral mucosa (6.00 ± 0.05 U/mg protein), and spleen (5.01 ± 0.05 U/mL), compared to the negative control (272.08 ± 3.43 U/mL, 8.00 ± 0.08 U/mg protein, and 5.72 ± 0.16 U/mg protein, respectively). The SOD enzymes of CMJ (0.5 g/kg)-treated groups remarkably activated to 289.16 ± 1.57 U/mL for the serum, 9.23 ± 0.07 U/mg protein for the oral mucosa, and 6.29 ± 0.06 U/mg protein for the spleen, compared to ulcer group. Increasing the CMJ dose to 1.0 g/kg increased the SOD activity of the serum and oral mucosa of the CMJ (1.0 g/kg)-treated groups. Meanwhile, the SOD of the spleen did not significantly change. The SOD of the serum and oral mucosa of the two positive groups significantly increased, but the spleen SOD activity was not significantly affected.

Oral ulcer induction caused a significant oxidative stress condition that represented as a significant rise in the H_2_O_2_ concentration of serum (35.15 ± 0.59 mM/L), oral mucosa (3.82 ± 0.13 mM/g), and spleen (1.80 ± 0.03 mM/g) in comparison to the corresponding values in the negative control (30.80 ± 0.32 mM/L, 1.42 ± 0.03 mM/g, and 0.655 ± 0.02 mM/g, respectively (*p* ≤ 0.001)), Table 4.

The H_2_O_2_ concentration of serum was significantly boosted as a response to CMJ force-feeding. Therefore, the H_2_O_2_ concentration of the serum of the CMJ (0.5 g/kg)-treated and CMJ (1.0 g/kg)-treated groups was higher than the H_2_O_2_ of the ulcer group by about 20.51 and 80.11%, respectively. An opposite trend was recorded in the H_2_O_2_ concentration of the oral mucosa and spleen of CMJ (0.5 g/kg)-treated and CMJ (1.0 g/kg)-treated groups; these were significantly reduced (about 35.86 and 29.85% for the oral mucosa and 73.89 and 77.78% for the spleen, respectively). Nystatin treatment caused a significant reduction in the H_2_O_2_ concentration of the serum, oral mucosa, and spleen compared to the ulcer group. CMJ force-feeding caused a substantial decrease in the H_2_O_2_ concentration of the serum, oral mucosa, and spleen of the positive groups compared to the negative group. Interestingly, CMJ behavior towards serum H_2_O_2_ production differed in healthy and infected animals. It significantly reduced the serum H_2_O_2_ in positive groups and also significantly raised the serum H_2_O_2_ in treated groups.

*Candida* infection significantly raised MDA levels, a lipid peroxidation biomarker, in the serum (6.32 ± 0.11 nmol/mL), oral mucosa (4.85 ± 0.08 nmol/g tissue), and spleen (3.53 ± 0.14 nmol/g tissue), compared to the negative group (*p* ≤ 0.001), Table 4. CMJ significantly reduced the MDA level of the serum, oral mucosa, and spleen in CMJ (0.5 g/kg)-treated and CMJ (1.0 g/kg)-treated groups compared to the ulcer control. No significant difference was noticed between the MDA levels (serum, oral mucosa, and spleen) of infected animals administered CMJ and the others administered nystatin. However, CMJ significantly decreased the MDA of the positive groups to levels lower than that of the negative group.

The NO concentration of *Candida* control was raised in the serum (28.73 ± 2.11 µmol/L), mucosal membrane (3.52 ± 0.25 µmol/g), and spleen (1.22 ± 0.07 µmol/g)), compared to the negative control, (19.77 ± 1.66 µmol/L), mucosal membrane (2.91 ± 0.24 µmol/g), and spleen (0.87 ± 0.06 µmol/g), respectively, (*p* ≤ 0.001). CMJ at two doses of 0.5 and 1.0 g/k also significantly increased the serum NO concentration of infected animals to 42.83 ± 1.06 and 57.60 ± 2.40 µmol/L, respectively, compared to *Candida* control (28.73 ± 2.11 µmol/L) (*p* ≤ 0.001). A high dose of CMJ increased serum NO concentration more than nystatin. CMJ at two doses remarkably maximized the serum NO of the positive control (32.97 ± 2.16 and 37.05 ± 2.02 µmol/L), in comparison to the negative control (19.77 ± 1.66 µmol/L) (*p* ≤ 0.001) Table 4.

Only a high dose of CMJ significantly raised the NO concentration in the mucosal membrane (5.94 ± 0.21 µmol/g) and spleen (1.41 ± 0.04 µmol/g), compared to *Candida* control (2.91 ± 0.24 and 0.87 ± 0.06 µmol/g, respectively) (*p* ≤ 0.001). CMJ at a high dose increased NO concentration in a manner similar to nystatin. In positive groups, CMJ did not affect NO concentration in the mucosal membrane and spleen, compared to the negative control.

### 2.3. Chemical Composition of CMJ

The CMJ (10.00 g extract/100 g powder) used had powder-like characteristics, with spinach-like odor and a light green color (Table 5). The total assessments carried out for CMJ exhibited that it contains polyphenols (66.33 ± 0.88 mg gallic acid/g extract), flavonoids (44.45 ± 1.45 mg quercetin /g extract), and tannins (14.49 ± 0.70 mg tannic acid/g extract); finally, alkaloids represented 200.43 ± 8.60 mg alkaloid/g extract.

### 2.4. Characterization of Phytoconstituents in Chenopodium murale Juice (CMJ) by HPLC/QTOF-HR-MS/MS Analysis

The valuable biological effects of CMJ prompted us to identify its phytochemical profile through a non-targeted profiling method using high-performance liquid chromatography (HPLC) coupled with a high-resolution quadrupole time-of-flight mass spectrometer (QTOF-MS) operated in both negative (−ve) and positive (+ve) ionization modes. CMJ was analyzed in (−ve) ionization mode.

Some 42 compounds belonging to different natural product classes are listed in Table 6; these were tentatively identified using HPLC-QTOF-HR-MS/MS, based on their retention times (R_t_), detected accurate mass, (−ve) ionization mode, and molecular formula; the error in ppm (between the observed mass and the real mass) of each phytochemical and the MS/MS fragment ions were used to determine the limit of detection for each peak of the compounds, by comparing the reference compounds’ spectra and reported data. The total ion current (TIC) and base peak (BPC) MS-chromatograms (Figure 2) revealed that the CMJ is rich in polyphenols such as phenolic acids, anthocyanines and flavonoids, especially kaempferol derivatives, aurone, flavones, flavonols and their glycosides.

The chemical structures of the individual polyphenolics were determined by analysis of fragment patterns, in which glycosides of flavonoids such as glucose, rhamnose, glucuronic acid and neohesperidoside (*m*/*z* 162, 146, 176 and 308) were cut out from their structures [17]. In this work, all of the separated compounds were tentatively identified and detected for the first time in CMJ. Some 13 compounds out of a total of 42 were already reported in *C. species* [18,19,20].

A total of 29 metabolites [2, 3, 5, (6–11), (13–15), 20, (22–24), 26, 29, (31–42)] were first detected in CMJ (Figure 2).

The relative concentration percentage (Rel. Conc.%) of the identified compounds was calculated, as shown in Table 6. It was found that the compounds of kaempferol derivatives, aurone, phenolics and fatty acids, anthocyanides, flavones, flavonols and their glycosides are the most concentrated compounds in the plant, compared to the rest of the separated compounds. It was noted specifically that the compound Kaempferol3-*O*-(6-p-coumaroyl)-glucoside has a high concentration that doubles compared to the rest of the high-concentration compounds.

LC-MS coupled with Q-TOF-MS also achieved the detection of chemical compounds with high sensitivity. The peaks of the most concentrated compounds were responsible for anti-candidal effect of CMJ, especially Kaempferol3-*O*-(6-p-coumaroyl)-glucoside, which is present in a concentration that significantly exceeds the rest of the Kaempferol derivatives. Different types of flavonoid glycosides in such abundance contribute to antifungal activity [21].
(1)$$Rel.Conc.(\%)=\frac{\mathrm{peak}\;\mathrm{area}\;(P.A)\mathrm{for}\;\mathrm{each}\;\mathrm{one}}{\mathrm{total}\;P.A}$$

#### 2.4.1. Characterization of the First Detected Kaempferol Derivatives

Flavonoid glycosides have a variety of isomers with the same molecular weight but distinct aglycone and sugar components at various places on the aglycone ring. It is difficult to determine the identification of the sugars and how they are connected using only mass spectrometry. For instance, the loss of 162 amu implies the presence of a hexose sugar, but it is unclear whether it is glucose or galactose. The locations and types of sugars were identified, as we compared the R_t_ values, mass spectra and chromatography of the main compounds, such as Kaempferol3-*O*-(6-p-coumaroyl)-glucoside, Kaempferol 3,7-*O*-bis-α-L-rhamnoside, Kaempferol3-*O*-glucoside, Luteolin-8-C-glucoside, Kaempferol3-*O*-α-L-rhamnoside(Afzelin), Maritimetin 6-*O*-glucoside, Kaempferol7-*O*-neohesperidoside, kaempferol3-*O*-robinoside-7-*O*-rhamnoside, and Kaempferol3-*O*-di-glucoside, under the same LC-MS conditions, with an authentic standard, and with the work of Zhuan-Hong, Li. et al. [22].

Seven kaempferol-*O*-glycosides were assigned for the first time in CMJ. Two characteristic molecular ion peaks in (−ve) mode at the same *m*/*z* (593.1433/593.1327) and different retention time (R_t_ 6.941 and 6.256 min) were detected as peaks 8 and 3.

Peak 8 gave product ions at *m*/*z* 447.1257[M − H-*p*-coumaroyl]^−^ and 431.1092[M − H-glucose(162)]^−^, and characteristic ions at *m*/*z* 307.000[M − H-Kaempferol (285)]^−^ and 285.0409[M − H-*p*-coumaroyl glucoside moiety (-308) amu)]^−^. The characteristic product ion at *m*/*z* 307 was attributed to the loss of *p*-coumaroyl group, rather than rhamnose moiety according to Karioti et al. [23]. Additionally, the loss of 308 Da (162 + 146) indicated that p-coumaroyl and a hexose linked at the same position of the aglycone. The substituent position of 3-OH was determined by the fact that the intensity of [M − H- (-308)]^−^ is higher than that of [M − H- (-308)]^−^, so compound 8 was proposed to be Kaempferol3-*O*-(6-*p*-coumaroyl)-glucoside [22,24].

Kaempferol3-O-(6-p-coumaroyl)-glucoside is formed by the acylation of the sugar moiety with hydroxycinnamic acid. Previous studies have shown that acylated flavonoid glycosides have higher antioxidant and antibacterial activities than their corresponding glycosides [23].

The product ions at *m*/*z* 447.0929[M − H-rhamnose]^−^, 431.0858[M − H-glucose]^−^ and 285.0331[M − H-(glucose + rhamnose)]^−^ were characteristic of the Kaempferol7-*O*-neohesperidoside; peak 3 due to the loss of rhamnose and glucose [25]. According to the loss of rhamnose(-146), the fragment at *m*/*z* 447.0929 proved that the deoxyhexose was in the terminal position. The loss of 308 Da (162 + 146) indicated that a hexose and a deoxyhexose were linked at the same position on the aglycone. The substitution of 7-OH glycosylation was proven by the predominant ion at *m*/*z* 285.0331[M − H-(308)]^−^, coupled with the weak ion at *m*/*z* 284.0239[M − H-(308)]^−^.

Peak 4 at R_t_ 6.506 min gave an [M − H]^−^ ion at *m*/*z* 739.2022. The characteristic fragmentation pathway of this compound in the MS/MS spectrum is based on the loss of different mono- and di-glycoside moieties at *m*/*z* 593.1429[M − H-rhamnose]^−^, 431.0969[M − H-(rhamnose+glucose) (308)]^−^ and 285.0380 [M − H-(308+rhamnose)]. Comparing the mass spectrum and chromatography with an authentic standard, compound **4** was tentatively established as kaempferol3-*O*-robinoside-7-*O*-rhamnoside [22].

Compound **5** showed a precursor ion at *m*/*z* 609.1412, and fragment ions at *m*/*z* 447.0953[M − H-glucose]^−^ and 285.0495[M − H-di glucose moieties (324)]^−^; it was therefore determined to be Kaempferol-3-*O*-di-glucoside. The substituent position of 3-OH was determined by the intensity of 284.0385[M − H-324]^−^, which is higher than that of 285.0495[M − H-324]^−^, and the loss of (-324) suggested the two hexoses were linked at the same position.

Two mono glycosides (peaks 2 and 9) of kaempferol had pseudomolecular ions at *m*/*z* 461.0726/431.0859, based on QTOF-MS/MS data reported in the literature. The compounds with product ions at 285.0389[M − H-lucuronic acid(176)]^−^, 284.9331[M − H-146)]^−^ and 151.0043 were characteristic of Kaempferol3-*O*-glucuronide and Kaempferol3-*O*-α-L-rhamnoside (Afzelin). Compounds **2** and **9** were further confirmed by comparing their mass spectra and chromatography with an authentic standard [26].

The metabolite 10 at (R_t_ 10.62 min) with the molecular formula C_16_H_12_O_6_ and a precursor ion at *m*/*z* 299.0561 was tentatively proposed to be 3, 5, 7-trihydroxy-4′-methoxyflavone (Kaempferide). This compound displayed a characteristic fragment ion at *m*/*z* 284.0330 due to the loss of [M − H-CH_3_]^−^. These results are completely in agreement with those reported in previous studies [27].

#### 2.4.2. Characterization of Flavone Derivatives

Observing HPLC-MS/MS spectral data, peaks 11, 13, 15 and 16 are mono-*O*-flavone glycosides; this could be determined from the fragmentation pattern of sugar units. Fragmentation due to the loss of 14 amu was detected for the methyl, the loss of 18 amu for the hydroxyl and the loss of 30 amu for the methoxy groups (Figure 3). Baicalein-7-*O*-glucuronide was detected for the first time, proposed for peak 11, with a molecular ion peak at *m*/*z* 445.0777, R_t_ 5.856 min, and a molecular formula of C_21_H_18_O_11_ [28].

Compound **13**, with a molecular ion peak at *m*/*z* 431.0980 and fragment ions at *m*/*z* 269.0482, 225.0428 and 151.0160, was identified as Apigenin7-*O*-glucoside. The intensity of 269.0482[M − H-162]^−^ coupled with the weak ion at *m*/*z* 268.0360[M − H-162]^−^ detected the substitution 7-OH glycosylation. In the same way, compound **16** was determined to be Luteolin7-*O*-glucoside due to the generation of a main ion at *m*/*z* 285.0411[M − H-162]^−^, which coupled with a weak ion at 284[M − H-162]^−^.

Compounds **14** and **15** were tentatively identified for the first time, and suggested to be 3′-methoxy-4′, 5, 7-trihydroxyflavone and Acacetin, with precursor ions at *m*/*z* 315.1239/283.0618 and a product ion at *m*/*z* 300.236/268.0374[M − H-CH_3_]^−^. LC-MS/MS spectral data showed a fragment ion at 327.0544[M − H-120] corresponding to the characteristic fragmentation pattern of 8-*C*-glycoside; therefore, it could be detected for the first time as Luteolin-8-*C*-glucoside, peak 12 [29].

#### 2.4.3. Characterization of the First Detected Flavanones and Aurone O-Glycosides

Two flavanone compounds **23** and **25** possessing molecular ions at *m*/*z* 593. 1327 and 271.0602 were characterized as Isosakuranetin-7-*O*-neohesperidoside, with characteristic product ions 446.0835 [M − H-rhamnose]^−^ and 285.0389 [M − H-(Rhamnose+glucose]^−^, and Naringenin [25]. The ion at *m*/*z* 446.0835 also proved that the deoxyhexose was in a terminal position. The loss of (−308 amu) showed the two sugars were linked at the same position. Compound **24**, with a molecular ion at *m*/*z* 447.0932, exhibited a fragmentation pattern similar to that of Maritimetin-6-*O*-glucoside, at *m*/*z* 285.0400, 151.0000 and 133.0000 [30]. Two intensive ions at *m*/*z* 285.0400 [M − H-162]^−^ and 284.0375[M − H-162]^−^ were observed in MS^2^, and proved that 6-*O*-glucoside was present in compound **24**.

#### 2.4.4. Characterization of Phenolics and Fatty Acids

Phenolic acids are a class of 2^ry^ metabolites (SM) that have a variety of interesting biochemical pathways. They usually form a pseudomolecular ion [M − H] corresponding to a deprotonated molecule and characteristic fragment ion [M − H-44] related to CO_2_ loss from the carboxylic acid group. In this work, six free phenolic acids were tentatively identified, including chlorogenic acid, quinic acid, caffeic acid, salicylic acid, P -hydroxybenzoic acid and 3,4-Dihydroxybenzoic acid [18,31], the highest concentration of which was found to be chlorogenic acid. Three highly concentrated fatty acids were detected as lactic acid, citraconic acid and D-3-Phenyllactic acid, respectively.

#### 2.4.5. Characterization of Flavonoids Aglycone

The systematic fragmentation of aglycones using LC-MS/MS experiments resulted in fragments at *m*/*z* 178.9976 and 151.0041 for quercetin; at 301.0313, 248. 9853, 125.0232 and 112.9899 for the first detected myricetin; at 151.0122, 133.0329 and 216.9352 for kaempferol; at 217.0430, 199.0415, 175.0441 and 151.0134 for luteolin; and at 151.0035, 117.0290 and 107.0172 for apigenin [32].

#### 2.4.6. Characterization of the First Detected Anthocyanins

The MS/MS spectrum of compound **22** with *m*/*z* 609.1418 [M−H]^−^ presents ions corresponding to *m*/*z* 447.0957 and 301.0355, which is in accordance with Delphinidin3-*O*-(6″-*O*-alpha rhamnopyranosyl *β*-glucopyranoside [33].

*Candida* yeasts are generally present in healthy humans, and are frequently part of the human body’s normal oral and intestinal flora, particularly on the skin; however, their growth is usually limited by the human immune system and by competition of other microorganisms. *Candida albicans* is not considered a pathogen in healthy individuals. However, in immunocompromised patients, it can cause severe systemic candidiasis [34]. In this study, a candidiasis model in rats was created as follows: (i) the rats were administered dexamethasone solution for 15 days to cause immunosuppression; (ii) the rats were infected with *Candida* for seven days; and (iii) the rats were treated with CMJ or nystatin for seven days. *Candida* control rats appeared to have a weak immune system, in which IL-17, IL-2, and INF-γ levels were statistically reduced. Additionally, altered hematological parameters, wherein rats recorded leukocytosis but the neutrophil% did not change significantly. Furthermore, *Candida* control rats suffered from oxidative stress, where antioxidant enzymes and CAT and SOD activity significantly decreased alongside significant elevation of oxidative stress biomarkers (i.e., H_2_O_2_ and MDA concentrations). Our results are in accordance with the immunosuppressive effect of glucocorticoid medications such as dexamethasone.

Dexamethasone is an immunosuppressive and anti-inflammatory glucocorticoid medication for treating many diseases, including (i) rheumatic problems, (ii) skin diseases, (iii) severe allergies, (iv) asthma, (v) chronic obstructive lung diseases, (vi) croup, (vii) brain swelling, (viii) eye pain following eye surgery, (ix) superior vena cava syndrome, and (x) antibiotics in tuberculosis. Unfortunately, dexamethasone has many side effects, including candidiasis and leukocytosis. Glucocorticoids work through inhibiting genes that code for the cytokines IL-1, IL-2, IL-3, IL-4, IL-5, IL-6, IL-8, and TNF-α, reducing T-cell proliferation. Glucocorticoids suppress hormonal immunity, causing B- cells to express fewer IL-2 and IL-2 receptors, diminishing both B cells’ clone expansion and antibody synthesis. In addition, glucocorticoids induce the resolution of inflammation by increasing the secretion of anti-inflammatory factors (IL-10 and tumor growth factor (TGF)-β). They also affect the adaptive immune system, suppressing CD4+ T cell activation by modulating dendritic cell function and promoting the polarization of T helper (Th) cells, with the preferential differentiation of Th2 and T regulatory (Treg) cells and the inhibition of Th1 and Th17 cells. In addition, glucocorticoids can alter patients’ microbiome and promote M2 macrophage polarization [35].

On the contrary, *Candida*-infected rats who received CMJ with two doses showed an opposite trend. CMJ caused an immunomodulatory action that manifested as a significant increase in INF-γ, IL-17, and IL-2. CMJ ameliorated hematological biomarkers toward phagocytosis, in which state the neutrophil% statistically increased. Meanwhile, the other percentages (lymphocyte, monocyte, basophil, and eosinophil) reduced considerably. CMJ activated antioxidant enzymes, CAT and SOD, and inhibited the oxidative stress biomarkers (i.e., H_2_O_2_ and NO) and MDA of the serum, mucosa, and spleen. CMJ showed a promising anti-fungal effect, immunomodulatory action, and antioxidant properties compared to nystatin treatment. CMJ showed the same trend as nystatin, an anti-fungal medication.

Both innate and adaptive immunity play a role in protection against fungal infections. Fungi are recognized by the innate immune system (e.g., dendritic cells and macrophages), resulting in phagocytosis and the initiation of killing mechanisms (e.g., production of reactive oxygen species) and helping to drive the development of adaptive immunity. In adaptive immunity, CD4+ T-cells cause IFN-γ (Th1) or IL-17 (Th17) to provide the best protection during fungal infections. This helps to drive effective killing by innate effector cells such as neutrophils and macrophages [36]. CMJ affected both innate and adaptive immunity, wherein it activated phagocytosis, which manifested as a significant elevation in neutrophils; it returned the INF-γ, IL-17, and IL-2 of the infected rats to a level close to that of the negative control.

Firstly, macrophages and neutrophils are the prototypical innate immune cell, essential for host defense against candidiasis. Neutrophils’ depletion in neutropenic patients leads to an increased susceptibility to mucosal candidiasis [37]. Neutrophils are highly phagocytic granulocytic polymorphonuclear cells that are important in antimicrobial immunity. A reduction in circulating neutrophils increases the risk of candidiasis [38]. Additionally, chronic granulomatous disease patients, whose neutrophils cannot make ROS, suffer from invasive aspergillosis [39]. The neutrophil’s classical killing mechanism works through the production of ROS and hydrolytic enzymes, which kill phagocytosed microbes when the granules that contain them fuse with the phagosome. The respiratory burst provides neutrophils with oxygen to generate ROS. The respiratory burst activates the NADPH oxidase, releasing large quantities of superoxide, a ROS [40]. Superoxide rapidly breaks down to H_2_O_2_ through SOD. Then, H_2_O_2_ is converted to hypochlorous acid (HClO) by myeloperoxidase. HClO is bactericidal enough to kill the bacteria phagocytosed by the neutrophil [41]. CMJ significantly elevated neutrophils and H_2_O_2_ and SOD activity, which suggests that CMJ promotes the classical phagocytosis of neutrophils by increasing ROS production.

Secondly, CMJ enhanced adaptive immunity, wherein it statistically elevated the levels of INF-γ, IL-17, and IL-2 that were significantly depleted in the *Candida* control.

INF-γ is a cytokine called the macrophage activation factor. It is produced by natural killer cells, dendritic cells, killer T cells, CD4 Th1, CD8 cytotoxic T lymphocytes, and effector T cells. INF-γ alters transcription in up 30 genes, producing a variety of physiological and cellular responses, including (i) promotion of natural killer cells, which indirectly fight *Candida* by promoting the production of TNF-α and INF-γ [37]; (ii) increasing antigen presentation and the lysosome effect of macrophages, causing them to engulf and digest the *Candida* cells (they are also essential for wound healing); (iii) activation of inducible NO synthase that mediates the cytotoxic effect of the pathogen; (iv) activation of B cells to produce IgG2a and IgG3 antibodies that enhance the neutrophil-mediated killing of *Candida*, opsonophagocytosis, and induce complement activation [36,42,43]; (v) induction of the expression of defense factors against retroviruses that have directly antiviral effects; and (vi) promotion of the adhesion and binding required for leukocyte migration [44,45].

IL-17 is a pro-inflammatory interleukin produced by CD4+ T cells. IL-17 is essential for protecting against mucosal candidiasis. The anti-candidal defense action of IL-17 was confirmed by the high susceptibility of ^−/−^ mice to mucosal candidiasis, which correlates with defects in neutrophil recruitment and reduced antimicrobial peptide (AMP) production. In addition, host recognition of *C. albicans* via immunoreceptors is required to mount an appropriate immune response, including activation of IL-17 [46]. Additionally, IL-17 inhibitory drugs used to treat psoriasis have been shown statistically to increase the risk of fungal and bacterial infections [47]. Finally, in humans, the *candida*-specific memory T cells are predominantly Th-17 cells [48]. IL-17 fights *Candida* infection through several mechanisms, including up-regulation of hematopoietic cells such as phagocytes, adaptive Th-17 cells, neutral Th-17 cells, dendritic cells, non-major histocompatibility complex-restricted T cells, and subsets γ and δ T cells. Additionally, IL-17 activates non-hematopoietic cells, including mucosal epithelial cells, which are responsible for expression of IL-17 receptors (IL-17RA/RC). In addition, IL-17 is involved in fungal clearance via the production of various pro-inflammatory cytokines and antimicrobial peptides during infection [33]. Th-17 produces cytokine IL-22, which is also vital for anti-fungal immune responses; however, in experimental OPC, IL-22^−/−^ mice are only mildly susceptible to *Candida* infection compared to IL-17RA^−/−^ mice [48]. IL-17 up-regulated many antimicrobial peptides’ excretion, including defensins (β-defensins), calprotectin (S100A8/9) and mucin [49]. IL-17 promotes expression of the mucin gene *MUC5B* that has implications for the inhibition of virulence factors, including genes of adhesion, filamentation and biofilm formation, which lead to *Candida* clearance [50]. IL-17 may be proven to affect immune modulation through the recruitment of additional IL-17+ lymphocytes to the site of infection. Signaling through the IL-17 receptor results in a neutrophil influx, which assists fungal control. Finally, Murine β-defensin-3, which is involved in protection against *Candida*, is strongly IL-17-dependent [51].

IL-2 is produced by activated CD4+ T cells and activated CD8+ T cells. IL-2 has several immunity functions, including (i) a direct effect on the antibody-producing clones of B lymphoblasts, and on activated T-lymphocyte proliferation; (ii) induction of the phagocytic capacity of polymorphonuclear leukocytes, and the destruction of target cells by natural killer cells and cytotoxic T lymphocytes; (iii) activation of non-major histocompatibility complex-restricted cytotoxic cells, resulting in the generation of lymphokine-activated killer cells with the capacity to kill tumor and fungal targets; (iv) lectin-like properties, which have specificity for high-mannose groups. Despite the previous essential functions of IL-2, it has mannose-binding properties, whereby IL-2 binds to *Candida* via its surface mannose groups. Cuneyt et al. [52] reported that Candida mannan has immunosuppressive properties. Lilic et al. [53] demonstrated that IL-2 and IL-10 cytokines are deregulated and *Candida*-specific. Incubation of *Candida albicans* with IL-2 inhibited its growth [54].

Finally, we suggest that the anti-fungal properties of CMJ may be due to the activation of CD4+ T cells, which causes significant increase in INF-γ, IL-2, and IL-17 production. Additionally, CMJ activated natural killer cells, dendritic cells, killer T cells, CD4+ Th1, CD8+ cytotoxic T lymphocyte, and effector T cells, which produce INF-γ.

Collectively, we may conclude that CMJ treated oral candidiasis in immunosuppressed rats via several strategies, including (i) promotion of classical phagocytosis of neutrophils; (ii) activation of T cells that activate IFN-γ, IL-2, and IL-17 production; (iii) increasing the production of cytotoxic NO and H_2_O_2_, which can kill *Candida*; (iv) activation of the SOD enzyme that converts superoxide to antimicrobial materials; and (v) induction of the repair process of damaged tissues. CMJ exhibited these characteristics due to its chemical composition; CMJ contains considerable quantities of polyphenols, flavonoids, and alkaloids. Additionally, LC-Mass identified several active constituents recognized in the CMJ, with anti-fungal and immunomodulatory effects.

For example, Kaempferol and kaempferol-bound gylcosides were the major components in the CMJ that were determined to be anti-fungal when tested alone or with the extract. Kaempferol-containing extracts such as *Scabiosa hymettia*, *Allium ursinum*, and *Bryophyllum pinnatum* exhibited antifungal activity against *Candida* types [55,56,57]. Pure kaempferol has a strong effect against oral and vaginal candidiasis [58,59,60]. Gallic acid suppressed the protein synthesis of *C. albicans*, reducing the number of hyphal cells and germ tubes [59]. Chlorogenic acid exhibited an in vitro anti-candidal effect, resulting in (i) cell viability reduction; (ii) the elevated possibility of mitochondria depolarization and the release of ROS; (iii) DNA fragmentation and phosphatidylserine liberation; and (iv) the induction of apoptosis. Additionally, chlorogenic acid exhibited considered interactions with the ALS3 active site residues of *C. albicans*, which enable it to adhere to and resist fluconazole [61]. Cinnamic acid has an in vitro immunoregulatory effect on monocytes against *C. albicans*. Caffeic acid inhibited *C. albicans* through an isocitrate lyase enzyme effect.

Quercetin and naringenin are good suppressors of *C. albicans*; they halted biofilm synthases and induced membrane disorders, thereby reducing cell size and infiltration of intracellular components. Quercetin activated phosphatidylserine, which inhibited fatty acid synthase, a main component of the cell wall. Quercetin regulated mitochondrial functions by (i) inhibiting oxidative phosphorylation; (ii) alternating the ROS production in mitochondria; and (iii) modulating the transcription factors that control mitochondrial proteins’ expression. The previous functions cause pro-apoptotic functions by discharging cytochrome C from the mitochondria; they can also do so indirectly by inducing the expression of the pro-apoptotic proteins of Bcl-2, and by reducing anti-apoptotic proteins. In vitro treatment with apigenin and quercitrin downregulated genes encoding efflux pumps (CDR1). Apigenin recorded antifungal activity against *C. albicans* through reducing the fungal virulence and expression of antifungal resistance-linked genes [62]. It also induces membrane disorders, leading to cell shrinkage and the loss of intracellular constituents [63,64]. Additionally, it causes mitochondrial disorders through stimulating mitochondrial calcium uptake, resulting in (i) membrane disruption; (ii) increased mitochondrial mass; and (iii) an elevation in ROS production. Furthermore, apigenin activated apoptosis via (i) phosphatidylserin exposure; (ii) DNA fragmentation; and (iii) caspase activation. Baicalein has potent antifungal activity against *C. albicans*, *C. tropicalis* and *C. parapsilosis* [65], wherein exposure of *Candida* cells to baicalin disrupted their biofilms’ components [66]. Myricetin has an antifungal effect against *C. albicans* via (i) injuring the cell wall’s integrity; (ii) increasing membrane permeability; (iii) causing DNA and protein loss; and (iv) causing alternations in the lipid components or order of the cell membrane [67].

## 3. Materials and Methods

### 3.1. Chemicals and Drugs

Folin–Ciocalteu reagent and authentic samples of gallic acid and quercetin were purchased from Sigma–Aldrich (St. Louis, MO, USA). Kits of SOD, CAT, MDA, and H_2_O_2_ were obtained from Biodigonestic Diagnostics Egypt (Dokki, Giza, Egypt). ELISA kits of Rat interleukin-17 and Rat interleukin-2 were obtained from Elabscience Biotechnology Inc. (Houston, TX 77079, USA). Meanwhile, an interferon-gamma (IFN-γ) rat ELISA kit was obtained from LSBio, LifeSpan BioSciences Inc. (Seattle, DC, USA, 98121). Tetracycline was obtained from Chemical Industries Development (CID) Company—Giza-A.R.E.-G.C.R. 19717. Dexamethasone sodium phosphate was obtained from AMRIYA Company (Alexandria, Egypt). Nystatin as a reference drug under trademark Fungistatin was obtained from the DELTA PHARMA S. A. E. Company (Ramadan City, El-Sharkia, Egypt). All of the used chemicals were of analytical grade.

### 3.2. Collection of C. murale and Preparation of the Juice

Leaves of *C. murale* were collected from fields in Sharkia Governorate, Egypt, in March 2021. Washed fresh leaves (500 g) were crushed using a blender (Toshiba, Egypt) and were filtrated by filter paper (Whatman No. 1). The filtrate was lyophilized using a lyophilizer. The residue extract (50 g) was kept at −20 °C until it was incorporated into the bioassay.

### 3.3. Anti-Candidiasis Activity of CMJ and Nystatin

#### 3.3.1. Assay of Acute Oral Toxicity (LD_50_)

The acute toxicity assay of the CMJ was performed according to according to per OECD guideline 423 (2001) for acute oral toxicity -Up-and-Down- Procedure (UDP) [68]. The dosing pattern started at 500 to 6000 mg/kg body weight. Mice were force-fed the extract by gastric tube (5 mice), and control mice received saline only. All groups were kept under observation and were assessed for any changes and mortality for 48 h. The live animals were observed for 14 days. Using mortality numbers in each concentration during the first 48 h and the BioStat program (BioStat 2009 Build 5.8.4.3^©^ 2023 analystSoft Inc., Alexandria, VA, USA), the CMJ LD_50_ was determined to be 5000 mg/kg.

#### 3.3.2. Microbial Strains and Culture Conditions

The *C. albicans* strain was isolated from the sputum culture of a patient suffering from oral candidiasis and was used for all experimental assays in this study. This strain was stored as frozen stocks in 30% glycerol at −80 °C, subcultured on Sabouraud Dextrose agar plates, and routinely grown in Sabouraud liquid medium at 37 °C. A single colony from the Sabouraud Dextrose agar plate was produced in a yeast extract–peptone glucose medium YPG: (yeast extract, 2%; bacto peptone, 1%; glucose, 2%) for 18 h at 30 °C in a shaker. The culture was harvested by centrifugation at 2500 rpm, and then cells were washed three times in phosphate-buffered saline (PBS) and adjusted to a final concentration of 5 × 10^6^ CFU/mL. Therefore, samples from the entire oral cavity or only from a specific site, e.g., the tongue’s surface, were collected with a swab, for posterior CFU counting, onto Sabouraud dextrose agar (SDA). After plating, the Petri dishes were incubated at 35–37 °C for 24 or 48 h (using a hemocytometer chamber for counting cells) [69].

#### 3.3.3. Animal Preparation and Oral Infection

Some 70 Wistar male albino rats (60 days old), weighing 180–200, were obtained from the Central Animal House of the National Research Centre, Giza, Egypt. The rats were maintained in cages at the animal care facility (20–25 °C, 55–65% humidity, 10–12 h light/dark cycle). The rats were fed the standard chow diet obtained from the Central Animal House. Water and food were available ad libitum over the experimental period.

#### 3.3.4. Oral Ulcer Induction

A model of oral candidiasis in immunosuppressed rats reported by Martinez et al. [4] was used. The experiment was performed in three stages: decreasing the immunity of the rats, infecting the rats with *Candida*, and treating them with CMJ. In the first stage, rats were immunosuppressed with dexamethasone and treated with tetracycline to enhance the infection rate. Before infection, for a week, the rats drank dexamethasone (0.5 mg/L) and tetracycline (0.1% to prevent bacterial infections) in water. On the day of infection, the rats drank dexamethasone (1 mg/L) and tetracycline (0.01%) in water. The rats were maintained with this water until the end of the experiment. In the second stage, the rats were orally infected three times at 48 h intervals (days-7, -5, and -3) with 0.1 mL of saline suspension containing 3.00 × 10^6^ viable cells of *C. albicans* (Table 7). Oral infection was achieved using a cotton swab rolled twice over all mouth parts. After the last inoculation (72 h), all groups were sampled using a cotton swab to confirm the infection and quantify the number of CFU (colony-forming units) in the oral cavity, before starting CMJ administration. Groups of mice were sacrificed under anesthesia and subjected to evaluation of the severity of lesions of the tongue. Macroscopic assessment of the infection was expressed by scoring lesions from 0 to 4 based on the extent and severity of whitish, curd-like patches on the tongue surface, as follows: 0, normal; 1, white patches in less than 20%; 2, white patches in less than 90% but more than 21%; 3, white patches in more than 91%; 4, thick white patches similar to pseudomembranes in more than 91%. In the third stage, rats were treated with CMJ or nystatin for seven days.

#### 3.3.5. The Dosing Protocol of CMJ and Nystatin

According to the LD_50_, the two doses of the extract were chosen to be 1.0 and 0.5 g/kg body weight, i.e., ¼ and 1/10 of LD_50_ [70,71]. Nystatin oral suspension was used as a standard drug. Nystatin was typically applied at the recommended dose of 1 mL/kg/day (equalling 100,000 units). CMJ and nystatin were dispersed in a viscous 0.8% agar solution as an excipient for oral treatment [72]. CMJ was typically applied using a 1 mL dropper similar to that of nystatin.

#### 3.3.6. Experimental Scheme

Rats were randomly divided into seven groups, every ten rats, as follows. Group I: rats received normal saline for 7 days and were maintained as a negative control. Group II and III: rats received CMJ at two doses, a high dose (1.0 g/kg/day) and a low dose (0.50 g/kg/day for 7 days), respectively, and were maintained as a positive control for these doses. Group IV: *Candida*-induced oral ulcer rats received normal saline for 7 days and were kept as an ulcer control. Group V and VI: *Candida*-induced oral ulcer rats were treated with CMJ at high doses (1.0 g/kg/day) and low doses (0.50 g/kg/day), respectively, for 7 days. Group VII: *Candida*-induced oral ulcer rats were treated with nystatin, the reference drug, at the recommended dose, for 7 days [72].

Some 48 h later, after the last drug administration, the rats were fasted overnight. The rats were anesthetized using an injection of ketamine 87 mg/kg of body weight and xylazine 13 mg/kg of body weight (dissolved in normal saline), and each rat received 0.2 mL/100 g body weight of this liquid [73]. Animals were sacrificed after anesthesia, and the blood samples were collected from the retro-orbital plexus and organs. Blood samples (2 mL) were collected over EDITA to CBC analysis. Blood samples were collected in a clean centrifuge tube and centrifuged at 3000 r/min for 10 min using Sigma Laborzentrifugen (Osterode am Harz, Germany). The organs were washed and weighted freshly. The oral mucosae were excised from the animals. The oral mucosae and spleens were homogenized using an ultrasonic tissue homogenizer, then centrifuged at 4000× *g*, at 4 °C for 15 min. Serum was used for determination of the antioxidant parameters and oxidative stress parameters. The oral mucosae and spleens were kept in 10% formalin for histopathological examination.

#### 3.3.7. Microbiological Evaluation of the Progression of the Infection

The whole oral cavity, including the buccal mucosa, tongue, soft palate, and other oral mucosal surfaces, was swabbed using a cotton pad. The end of the cotton pad was then cut off and placed in a tube containing 5 mL sterile saline. After mixing on a Vortex mixer to release *Candida* cells from the swab into the saline, serial 100-fold dilutions of the cell suspension were incubated on a *Candida* GS plate at 37 °C for 20 h. The CFU (colony-forming units) of the *Candida* colonies were counted.

#### 3.3.8. Biochemical Analysis

In accordance with the work of Dacie and Lewis [74], blood samples were introduced to determine the hematological parameters. In addition, antioxidant biomarkers and catalase (CAT) and superoxide dismutase (SOD) activities of the serum, mucosa, and spleen tissues were determined spectrophotometrically [75,76].

The oxidative stress biomarkers, malondialdehyde (MDA), nitric oxide (NO), and hydrogen peroxide (H_2_O_2_), of the serum, mucosa, and spleen homogenate were spectrophotometrically estimated [77,78,79].

According to the manufacturer’s instructions, using an enzyme-linked immunosorbent assay, serum interleukins, IL-17 and IL-2, and interferon-gamma (IFN-γ) were determined using ELISA kits.

### 3.4. Chemical Composition of the CMJ

#### Quantitative Analysis of the CMJ

The total phenolic content was estimated using the Folin–Ciocalteu method [80]. The total flavonoid content was measured according to Lin and Tang [81], and was expressed as mg quercetin/g of extract. Crude alkaloids were gravimetrically determined according to Onwuka [82]. The tannin content was evaluated by the standard method of Broadhurst and Jones [83].

### 3.5. Qualitative Analysis of Phytoconstituents in CMJ by HPLC/QTOF-HR-MS/MS

Liquid chromatography-mass/mass spectrometry analysis was used to identify the chemical composition of CMJ [84,85]. LC-mass/mass analysis was carried out in the Proteomics and Metabolomics Research Program of the Basic Research Department at the Children’s Cancer Hospital, Cairo, Egypt.

#### 3.5.1. Sample Preparation

A stock solution of the CMJ was prepared from 50 mg of the lyophilized CMJ dissolved in 1000 µL of the solvent mixture, itself composed of water: methanol: acetonitrile (H_2_O:MeOH:ACN) in a ratio of 2:1:1. Complete solubility of the stock solution was obtained by vortexing the sample for 2 min and ultra-sonicating the sample at 30 kHz for 10 min. An aliquot, 20 µL of the stock solution, was again diluted with 1000 µL of H_2_O:MeOH:ACN (2:1:1) and centrifuged at 10,000 rpm for 10 min. Finally, 10 µL of stock with a concentration of 2.5 µg/µL was injected. Some 10 µL of reconstitution solvent was injected as a blank sample. The sample was injected in negative modes.

#### 3.5.2. Instruments and Acquisition Method

The mass spectrometry (MS) was performed on a Triple TOF 5600+ system equipped with a duo-spray source operating in the ESI mode (AB SCIEX, Concord, ON, Canada). The sprayer capillary and declustering potential voltages were −4500 and −80 V in the negative mode. The source temperature was set at 600 °C, the curtain gas was 25 psi, and gas 1 and gas 2 were 40 psi. A collision energy of −35 V (negative mode), a CE spreading of 20 V and an ion tolerance of 10 ppm were used. The TripleTOF5600+ was operated using an information-dependent acquisition (IDA) protocol. Batches for MS and MS/MS data collection were created using Analyst-TF 1.7.1. The IDA method was used to simultaneously collect full-scan MS and MS/MS information. The technique consisted of high-resolution survey spectra from 50 to 1100 *m*/*z*, and the mass spectrometer was operated in a pattern, wherein a 50-ms survey scan was detected. Subsequently, after each scan, the top 15 intense ions were selected for acquisition of the MS/MS fragmentation spectra.

#### 3.5.3. LC-MS Data Processing

MS-DIAL 3.70 open-source software was used for the sample’s non-targeting, small molecule comprehensive analysis. ReSpect negative (1573 records) databases were used as reference databases according to the acquisition mode. The MS-DIAL output was used to run again on PeakView 2.2 with the Master View 1.1 package (AB SCIEX) for feature (peaks) confirmation, based on the criteria, using the total ion chromatogram (TIC). Aligned features had a signal-to-noise ratio greater than 5, and a sample intensity of greater than 5.

### 3.6. Statistical Analysis

Data were analyzed as mean ± SE for every six rats. Comparisons among groups were performed using a one-way analysis of variance ANOVA test, at *p* ≤ 0.001, followed by a Tukey comparison test using IBM-SPSS (version 25), followed by a post hoc test.

## 4. Conclusions

The current study demonstrated the antifungal effect of *C. murale* fresh juice (CMJ). CMJ treated oral candidiasis in immunosuppressed rats, where the *Candida* count in mucosal parts was dramatically reduced. Concurrently, CMJ exhibited significant enhancement of neutrophils, NO, H_2_O_2_ production and SOD activity, meaning CMJ fought *Candida* through classical phagocytosis of the neutrophils by increasing ROS production. Besides, CMJ promoted adaptive immunity and significantly increased INF-γ, IL-17, and IL-2 production. The antifungal and immunomodulatory effects of CMJ may attributed to its active constituents, which were identified by LC-MS/MS. This study can be generalized to a broader study population, and may prompt a clinical trial using CMJ as an antifungal drug.

## Figures and Tables

**Figure 1 molecules-28-04304-f001:**
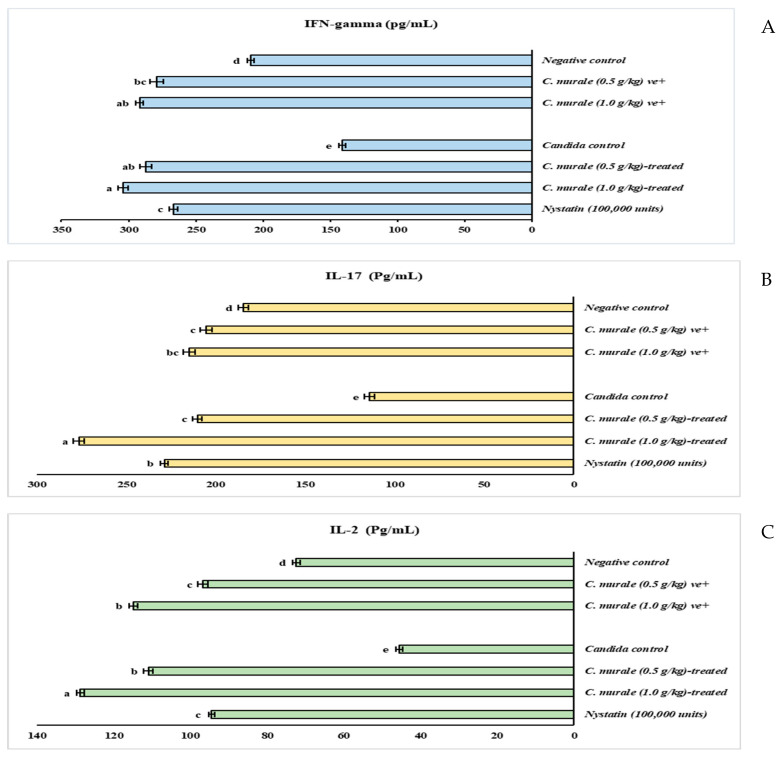
The effect of *Chenopodium murale* juice (CMJ) on the INF-gamma (**A**), IL-17 (**B**), and IL-2 (**C**) of oral candidiasis in immunosuppressed rats. Data are presented as mean ± SE. Data were analyzed using a one-way ANOVA followed by post hoc analysis for multiple comparisons, and *p* ≤ 0.001. Value with the different superscript letters means significance at a probability level of 0.1%.

**Figure 2 molecules-28-04304-f002:**
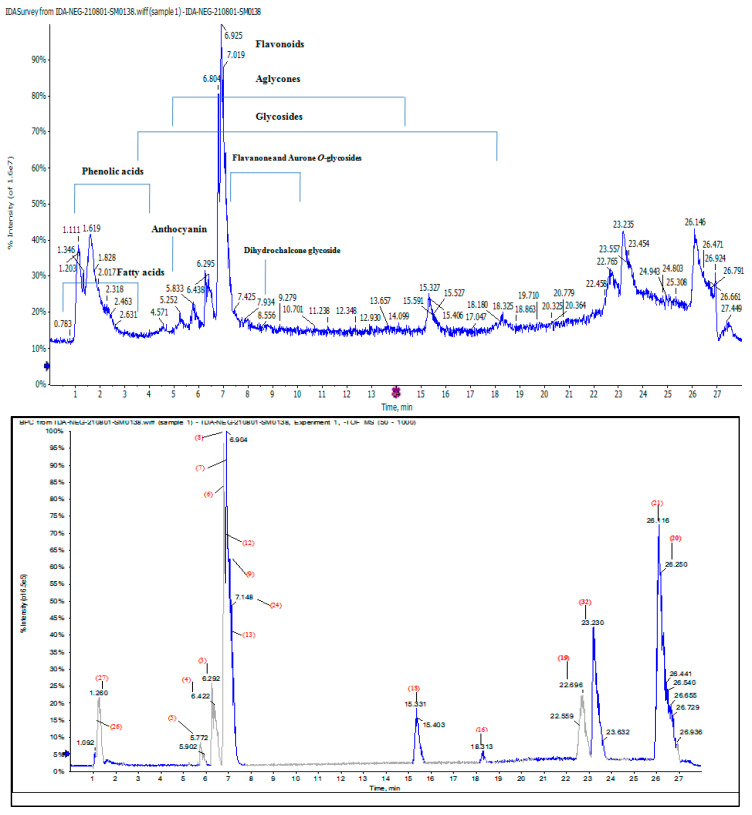
TIC and BPC chromatograms of *Chenopodium murale* juice (CMJ) using LC-QTOF-HR-MS/MS in negative ionization mode (−ve).

**Figure 3 molecules-28-04304-f003:**
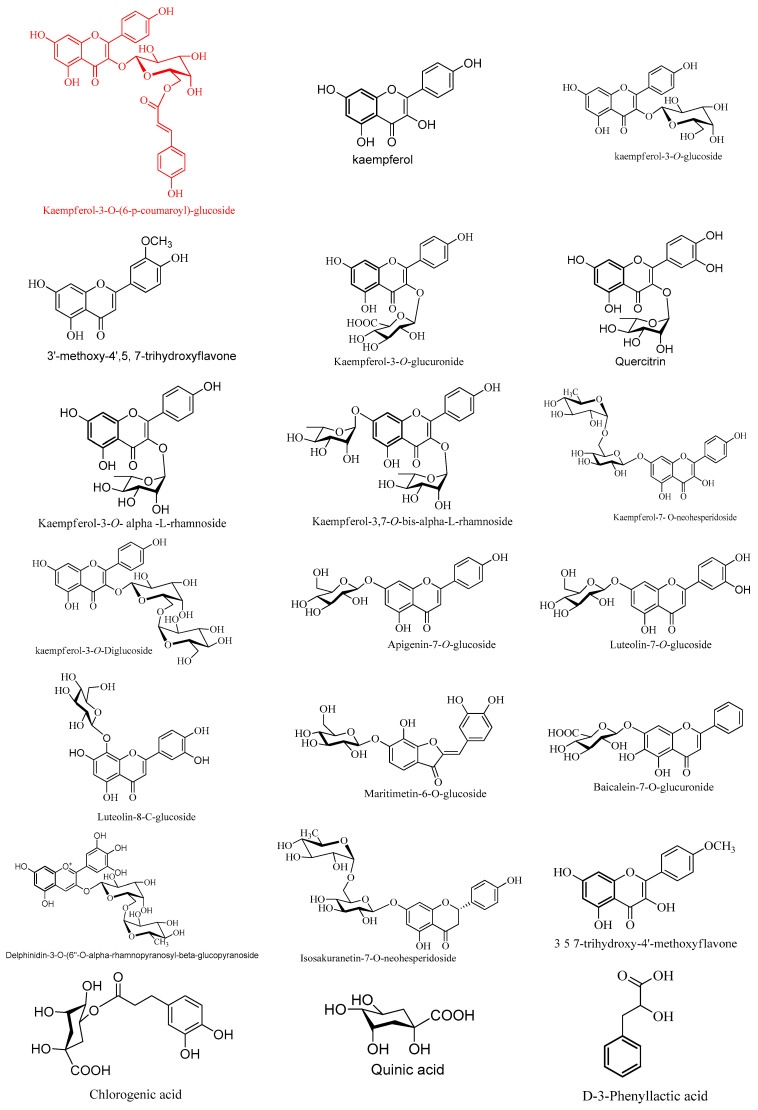
Structural skeletons of the most concentrated constituents of *Chenopodium murale* juice (CMJ), found by HPLC/QTOF.

**Table 1 molecules-28-04304-t001:** Microbiological study of therapeutic efficacy of *Chenopodium murale* juice (CMJ) against oral candidiasis in immunosuppressed rats.

Groups	Strength	CFU/Petri
Negative control	+ve	11.33 ± 1.53
*C. murale* (0.5 g/kg) ve+	−ve	1.00 ± 0.09
*C. murale* (1.0 g/kg) ve+	−ve	0.87 ± 0.07
*Candida* control	+++	5.86 × 10^4^ ± 1.21
*C. murale* (0.5 g/kg)-treated	++	236.67 ± 37.86
*C. murale* (1.0 g/kg)-treated	+	4.33 ± 0.58
Nystatin (100,000 units)	−ve	1.67 ± 0.29

Data are presented as mean ± SE. Data were analyzed using a one-way ANOVA followed by post hoc analysis for multiple comparisons, and *p* < 0.001.

**Table 2 molecules-28-04304-t002:** The effect of *Chenopodium murale* juice (CMJ) on the CBC of oral candidiasis in immunosuppressed rats.

Groups	Hemoglobin (g/dL)	RBC (×10^12^/L)	WBC (×10^9^/L)	Platelets (×10^9^/L)	Lymphocyte %	Neutrophil %	Monocytes %	Eosinophil %	Basophil %
Negative control	11.87 ± 0.16 ^ab^	5.89 ± 0.06 ^a^	8.60 ± 0.38 ^b^	436.50 ± 6.43 ^b^	63.60 ± 0.29 ^a^	27.25 ± 1.14 ^d^	6.07 ± 0.55 ^d^	2.69 ± 0.24 ^cd^	0.40 ± 0.04 ^c^
*C. murale* (0.5 g/kg) ve+	12.83 ± 0.32 ^ab^	6.09 ± 0.16 ^ab^	6.97 ± 0.18 ^d^	448.35 ± 15.21 ^b^	59.17 ± 3.67 ^b^	30.73 ± 2.29 ^c^	6.99 ± 0.78 ^c^	2.66 ± 0.30 ^cd^	0.44 ± 0.05 ^c^
*C. murale* (1.0 g/kg) ve+	11.80 ± 0.24 ^ab^	5.93 ± 0.03 ^ab^	7.23 ± 0.44 ^d^	525.00 ± 1.48 ^a^	54.99 ± 0.81 ^c^	35.07 ± 1.01 ^a^	6.88 ± 0.51 ^c^	2.62 ± 0.19 ^d^	0.43 ± 0.03 ^c^
*Candida* control	12.68 ± 0.96 ^ab^	5.43 ± 0.34 ^b^	13.30 ± 1.50 ^a^	194.75 ± 5.14 ^e^	59.07 ± 4.22 ^b^	26.50 ± 2.44 ^d^	9.56 ± 0.79 ^a^	4.24 ± 0.35 ^a^	0.63 ± 0.05 ^a^
*C. murale* (0.5 g/kg)-treated	10.13 ± 0.29 ^b^	5.19 ± 0.07 ^b^	4.50 ± 0.23 ^e^	234.99 ± 5.06 ^d^	55.65 ± 0.17 ^c^	32.92 ± 1.92 ^b^	8.03 ± 0.90 ^b^	2.90 ± 0.33 ^c^	0.50 ± 0.06 ^b^
*C. murale* (1.0 g/kg)-treated	12.70 ± 0.15 ^ab^	5.98 ± 0.12 ^ab^	4.77 ± 0.64 ^e^	321.98 ± 8.33 ^c^	55.18 ± 0.66 ^c^	35.68 ± 1.77 ^a^	6.42 ± 0.59 ^cd^	2.32 ± 0.21 ^e^	0.40 ± 0.04 ^c^
Nystatin (100,000 units)	13.33 ± 0.59 ^a^	6.43 ± 0.18 ^a^	7.85 ± 0.74 ^c^	320.76 ± 2.72 ^c^	57.17 ± 3.97 ^bc^	31.27 ± 2.55 ^bc^	7.66 ± 0.68 ^b^	3.40 ± 0.37 ^b^	0.51 ± 0.05 ^b^

Data are presented as mean ± SE. Data were analyzed using a one-way ANOVA followed by post hoc analysis for multiple comparisons, and *p* < 0.001. The means followed by the same letter in each column are not significantly different at the 0.1% probability level (Duncan’s multiple range test).

**Table 3 molecules-28-04304-t003:** The effect of *Chenopodium murale* juice (CMJ) on the CAT and SOD (serum, mucosa, and spleen) of oral candidiasis in immunosuppressed rats.

Groups	Catalase (CAT)			Superoxide Dismutase (SOD)		
	Serum (U/L).	Mucosa (U/mg Protein)	Spleen (U/mg Protein)	Serum (U/mL).	Mucosa (U/mg Protein)	Spleen (U/mg Protein)
Negative control	250.42 ± 2.02 ^f^	12.98 ± 0.58 ^d^	12.77 ± 0.42 ^b^	272.08 ± 3.43 ^c^	8.00 ± 0.08 ^c^	5.72 ± 0.16 ^bc^
*C. murale* (0.5 g/kg) ve+	353.92 ± 3.17 ^e^	12.76 ± 0.18 ^d^	13.79 ± 1.03 ^b^	287.18 ± 3.43 ^ab^	8.83 ± 0.07 ^bc^	6.09 ± 0.16 ^b^
*C. murale* (1.0 g/kg) ve+	379.85 ± 2.85 ^d^	14.29 ± 0.33 ^c^	19.16 ± 0.39 ^a^	300.98 ± 0.48 ^a^	13.73 ± 0.36 ^a^	6.14 ± 0.36 ^b^
*Candida* control	202.43 ± 8.80 ^g^	7.67 ± 0.39 ^e^	12.74 ± 0.16 ^b^	226.52 ± 3.94 ^d^	6.00 ± 0.05 ^d^	5.01 ± 0.05 ^c^
*C. murale* (0.5 g/kg)-treated	498.50 ± 4.30 ^b^	17.16 ± 0.19 ^ab^	15.81 ± 0.47 ^b^	289.16 ± 1.57 ^ab^	9.23 ± 0.07 ^b^	6.29 ± 0.06 ^b^
*C. murale* (1.0 g/kg)-treated	550.86 ± 3.61 ^a^	18.09 ± 0.80 ^a^	20.87 ± 1.19 ^a^	297.55 ± 2.27 ^a^	13.88 ± 0.16 ^a^	5.79 ± 0.10 ^bc^
Nystatin (100,000 units)	406.36 ± 4.53 ^c^	15.75 ± 0.19 ^bc^	13.50 ± 0.32 ^b^	281.37 ± 1.98 ^bc^	8.81 ± 0.17 ^bc^	7.49 ± 0.30 ^a^

Data are presented as mean ± SE. Data were analyzed using a one-way ANOVA followed by post hoc analysis for multiple comparisons, and *p* ≤ 0.001. The means followed by the same letter in each column are not significantly different at the 0.1% probability level (Duncan’s multiple range test).

**Table 4 molecules-28-04304-t004:** The effect of *Chenopodium murale* juice (CMJ) on the oxidative stress biomarkers (serum, mucosa, and spleen) of oral candidiasis in immunosuppressed rats.

Groups	Hydrogen Peroxide (H_2_O_2_)			Malondialdehyde (MDA)			Nitric Oxide Concentration (NO)		
	Serum (mM/L)	Mucosa (mM/g)	Spleen (mM/g)	Serum (nmol/mL)	Spleen (nmol/g)	Mucosa (nmol/g)	Serum (µmol/L)	Mucosa (µmol/g)	Spleen (µmol/g)
Negative control	30.80 ± 0.32 ^d^	1.42 ± 0.03 ^d^	0.655 ± 0.02 ^b^	1.88 ± 0.06 ^b^	1.17 ± 0.04 ^bc^	2.52 ± 0.08 ^b^	19.77 ± 1.66 ^f^	2.91 ± 0.24 ^c^	0.87 ± 0.06 ^d^
*C. murale* (0.5 g/kg) ve+	27.11 ± 0.60 ^e^	1.44 ± 0.04 ^d^	0.40 ± 0.01 ^de^	1.56 ± 0.04 ^bc^	0.95 ± 0.03 ^d^	1.95 ± 0.05 ^d^	32.97 ± 2.16 ^d^	3.19 ± 0.52 ^bc^	0.73 ± 0.04 ^e^
*C. murale* (1.0 g/kg) ve+	26.45 ± 0.59 ^e^	1.34 ± 0.03 ^d^	0.37 ± 0.01 ^e^	1.47 ± 0.03 ^c^	0.98 ± 0.02 ^cd^	1.77 ± 0.04 ^c^	37.05 ± 2.02 ^c^	3.06 ± 0.12 ^bc^	0.82 ± 0.03 ^de^
*Candida* control	35.15 ± 0.93 ^c^	3.82 ± 0.13 ^a^	1.80 ± 0.03 ^a^	6.32 ± 0.11 ^a^	4.85 ± 0.08 ^a^	3.53 ± 0.14 ^a^	28.73 ± 2.11 ^e^	3.52 ± 0.25 ^c^	1.22 ± 0.07 ^b^
*C. murale* (0.5 g/kg)-treated	42.359 ± 0.61 ^b^	2.45 ± 0.08 ^bc^	0.47 ± 0.02 ^cd^	1.75 ± 0.09 ^bc^	1.23 ± 0.06 ^b^	2.45 ± 0.12 ^b^	42.843 ± 1.06 ^b^	3.38 ± 0.2 ^bc^	1.09 ± 0.02 ^c^
*C. murale* (1.0 g/kg)-treated	63.31 ± 0.92 ^a^	2.68 ± 0.04 ^b^	0.40 ± 0.02 ^de^	1.80 ± 0.03 ^b^	1.22 ± 0.02 ^b^	2.48 ± 0.05 ^b^	57.60 ± 2.40 ^a^	5.94 ± 0.21 ^a^	1.41 ± 0.04 ^a^
Nystatin (100,000 units)	26.51 ± 0.36 ^e^	2.22 ± 0.11 ^c^	0.53 ± 0.01 ^c^	1.75 ± 0.02 ^bc^	1.31 ± 0.02 ^b^	2.45 ± 0.03 ^b^	44.66 ± 1.83 ^b^	5.55 ± 0.32 ^a^	1.47 ± 0.11 ^a^

Data are presented as mean ± SE. Data were analyzed using a one-way ANOVA followed by post hoc analysis for multiple comparisons, and *p* ≤ 0.001. The means followed by the same letter in each column are not significantly different at the 0.1% probability level (Duncan’s multiple range test).

**Table 5 molecules-28-04304-t005:** Physical examination and organoleptic characters of the *Chenopodium murale* juice (CMJ).

Yield and Physical Characters	Ethanol 70%
Yield (g, %)	10.00 g extract/100 g dried powder
Color	Light green
Odor	Like-spinach
Condition	powder
Phenols	66.33 ± 0.88 mg gallic acid/g extract
Flavonoids	44.45 ± 1.45 mg quercetin /g extract
Tannins	14.49 ± 0.70 mg tannic acid/g extract
Alkaloids	200.43 ± 8.60 mg alkaloid/g extract

**Table 6 molecules-28-04304-t006:** Metabolites identified in *Chenopodium murale* juice (CMJ) using LC-QTOF-MS/MS in negative ionization mode (−ve).

No	RT(min)	Name	Class	[M − H]^−^ *m*/*z*	Diff (ppm)	MF	MS^2^	Rel. Conc. (%)
**Flavonol glycosides**								
1	5.073	Quercetin 3-*O*-rhamnoside (Quercitrin)	Flavonol mono-glycosides	447.0901	−1	C_21_H_20_O_11_	301.0344, 273.0396, 151.0023	0.07528807
2	5.230	Kaempferol-3-*O*-glucuronide	Flavonol glycosides	461.0726	0.7	C_21_H_18_O_12_	285.0389	0.020850167
3	6.256	Kaempferol-7-*O*-neohesperidoside	Flavonol di-glycosides	593.1327		C_27_H_30_O_15_	431.0858 285.0331	0.34005973 *
4	6.506	kaempferol-3-*O*-robinoside-7-*O*-rhamnoside	Flavonol glycosides	739.2022	1.7	C_33_H_40_O_19_	593.1429, 431.0969, 285.0380	0.26219089 *
5	5.559	Kaempferol-3-*O*-di-glucoside	Flavonol di-glycosides	609.1412	-	C_27_H_30_O_16_	447.0953, 285.0495	0.32304453 *
6	6.865	Kaempferol-3-*O*-glucoside	Flavonol glycosides	447.0907		C_21_H_20_O_11_	285.0373, 284.0337	0.19927864 *
7	6.902	Kaempferol-3,7-*O*-bis-α-L-rhamnoside	Flavonol mono glycosides	577.01543	0.7	C_27_H_30_O_14_	431.0969, 285.0405	0.24868134 *
8	6.941	Kaempferol-3-*O*-(6-p-coumaroyl)-glucoside	Flavonol glycosides	593.1433	1.8	C_30_H_26_O_13_	447.1257, 431.1092, 307.000, 285.0409	1.49217077 **
9	6.973	Kaempferol-3-*O*-α-L-rhamnoside (Afzelin)	Flavonol mono-glycosides	431.0859	−1	C_21_H_19_O_10_	284.9331, 255.0280, 227.0275	0.21337071 *
10	10.062	3,5,7-trihydroxy-4′-methoxyflavone (Kaempferide)	methoxy flavonol	299.0561	−2.5	C_16_H_12_O_6_	284.0330,253.2070, 271.0268, 183.488	0.04408804
**Flavones glycosides**								
11	5.856	Baicalein-7-*O*-glucuronide	Flavone	445.0777	0	C_21_H_18_O_11_	269.0482, 175.0253, 113.0260	0.06855429
12	6.939	Luteolin-8-C-glucoside	Flavone	447.0964		C_21_H_20_O_11_	327.0544, 285.0401	0.39543838 *
13	7.730	Apigenin-7-*O*-glucoside	Flavone	431.0980	00	C_21_H_20_O_10_	269.0482, 225.0428, 151.0160	0.1863942 *
14	12.590	3′-methoxy-4′,5, 7-trihydroxyflavonol	Flavone	315.1239		C_16_H_12_O_6_	300.236, 246.8960, 163.0411	0.264702 *
15	14.036	Acacetin	4′-*O*-methylated flavone	283.0618	5.8	C_16_H_12_O_5_	268.0374, 252.0737, 151, 171, 211, 240, 239	0.02014899
16	18.155	Luteolin-7-*O*-glucoside	Flavone	447.887	−1.7	C_21_H_20_O_11_	285.0400, 242.9547,151.05012	0.17331039 *
**Aglycones**								
17	9.254	Luteolin	Flavone	285.415	−28.6	C_15_H_10_O_6_	217.0430, 199.0415, 175.0441, 151.0134	0.04887441
18	15.073	Kaempferol	Flavonol	285.0401		C_15_H_10_O_6_	151.0122,133.0329, 216.9352	0.18664631 *
19	22.721	Quercetin	Flavonol	301.328	0.5	C_15_H_10_O_7_	255.2321, 178.9976,151.0041, 121.0311	0.171138974 *
20	26.325	Myricetin	Flavonol	316.1284	7.4	C_15_H_10_O_8_	301.0313, 248.9645, 151.00, 112.9864	0.2735515 *
21	26.168	Apigenin	Flavone	269.0440	33.3	C_15_H_10_O_5_	151.0035, 117.0290, 107.0172	0.252827 *
**Anthocyanin, flavanones and Aurone *O*-glycosides**								
22	5.750	Delphinidin-3-*O*-(6″-*O*-alpha-rhamnopyranosyl-*β*-glucopyranoside	Anthocyanin	609.1418	40.3	C_27_H_31_O^+^_16_	447.0959, 463.0886, 301.0399, 299.0234	0.09779782 *
23	9.390	Isosakuranetin-7-*O*-neohesperidoside (Poncirin)	Flavanones	593. 1327	1	C_28_H_34_O_14_	447.0835, 285.0331	0.01449699
24	7.180	Maritimetin-6-*O*-glucoside	Flavonoid (Aurone *O*-glycosides)	447.0932	−0.7	C_21_H_20_O_11_	285.0400, 151.0000, 133.0000	0.44026256 *
25	10.001	Naringenin	Flavanones	271.0602	−1.2	C_15_H_12_O_5_	151.0032, 177.071	0.06571804
**Phenolic acids**								
26	1.129	Quinic acid	Phenolic acid	191.0549	16.4	C_7_H_12_O_6_	173.0448, 109.0276	0.2669481 *
27	1.259	Chlorogenic acid	Phenolic acid	353.0887	2.7	C_16_H_18_O_9_	191.0557, 173.0461	0.38734244 *
28	2.257	Caffeic acid	Phenolic acid	179.0276	−0.3	C_9_H_8_O_4_	135.0455	0.03597749
29	2.953	Salicylic acid	Phenolic acid	137.035	−0.9	C_7_H_6_O_3_	93.0339	0.04471586
30	4.173	P-Hydroxybenzoic acid	Phenolic acid	137.0258	−0.5	C_7_H_6_O_3_	119.0111, 93.0000	0.02646657
31	4.559	3,4-Dihydroxybenzoic acid	Phenolic acid	153.0199	1.7	C_7_H_6_O_4_	135.0092, 109.0300	0.21523038
**Fatty acids**								
32	23. 293	Lactic acid	Fatty acids	89.0245	2.3	C_3_H_6_O_3_	Not fragment	0.48174368 *
33	1.522	Citraconic acid	Fatty acids	128.9882	1.9	C_5_H_6_O_4_	Not fragment	0.3839927 *
34	1.91	D-3-Phenyllactic acid	Fatty acids	165.0403	2.8	C_9_H_10_O_3_	147.0330	0.11360878 *
35	2.252	Glutaric acid	Fatty acids	131.0695	−7.3	C_5_H_8_O_4_	86.9960	0.00505673
36	2.271	Suberic acid	Fatty acids	173.1171	0.2	C_8_H_14_O_4_	Not fragment	0.01325589
**Sugars**								
37	1.045	L-(+)-Tartrate	Sugars	148.9536	−1.7	C_4_H_6_O_6_	Not fragment	0.02088193
38	4.0873	D-(+)-Raffinose	Oligosaccharides	503.0786	−2.8	C_18_H_32_O_16_	Not fragment	0.00989468
39	8.792	D-(+)-Galacturonic acid	Sugars	193.0507	−7.7	C_6_H_10_O_7_	133.0303, 121.0302	0.12838032 *
**Organic compounds and dihydrochalcone glucosides**								
40	3.210	Xanthine	Xanthinones	151.387		C_5_H_4_N_4_O_2_	107.418	0.00863697
41	7.775	P-Nitro phenol	phenols	138.202		C_6_H_5_NO_3_	108.0224, 92.02	0.01166895
42	7.996	Phlorizin	dihydrochalcone glucosides	435.1151		C_21_H_24_O_10_	273.0760	0.00015185

Relative concentration (%) calculated based on the individual peak areas for identified components, * highly concentrated compounds; ** The most concentrated compounds.

**Table 7 molecules-28-04304-t007:** Experimental layouts.

Stages	Experimental Days	Protocols
Adaptation period	1st–7th days	Animals were kept in laboratory conditions. water and food provided ‘*ad labium*’.
Immunosuppression stage	8th–14th days	Animals received dexamethasone (0.5 mg/ L) and 0.1% tetracycline in drinking water
Infection stage	15th day	Animals were infected with *Candida* albicans suspension (3.00 × 10^6^ viable cell/ mL). Dexamethasone was increased to 1 mg/L Tetracycline was decreased to 0.01%
1st insemination	16th day	
	17th day	
2nd insemination	18th day	
	19th day	
3rd insemination	20th day	
1st Sampling for *Candida* detection	In day 21	
Treatment stage	In day 22	Animals were treated with *Chenopodium* juice (0.5 or 1.0 g/kg/day) or nystatin (1000.000 IU/mL) Dexamethasone and tetracycline were stopped
	In day 23	
	In day 24	
	In day 25	
	In day 26	
	In day 27	
	In day 28	
2nd Sampling for *Candida* detection	In day 29	The end of experiment

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
