# Peer review of "Chenopodium murale Juice Shows Anti-Fungal Efficacy in Experimental Oral Candidiasis in Immunosuppressed Rats in Relation to Its Chemical Profile"

_molecules, 2023, doi:10.3390/molecules28114304_

Round 1
Reviewer 1 Report (Previous Reviewer 1)
This manuscript reports anti-fungal efficacy in experimental oral candidiasis in immunosuppressed rats in relation to chemical profile. Therefore the identification of the constituents is very important.
The constituents have been identified based on only LC-QTOF-MS/MS data without comparing with data of authentic samples. Without direct comparison with authentic samples, the location of sugar unit can not be determined.
Thus, this manuscript does not contain sufficient data, which means that "Major revision" or "Reject" is adequate for this manuscript.
Author Response
Author's Response
Responses to reviewers’ comments for “Chenopodiastrum murale juice shows anti-fungal efficacy in experimental oral candidiasis in immunosuppressed rats” (Manuscript ID: molecules-2281968)
Dear Reviewer 1,
We are very pleased to receive your letter containing the reviewers’ comments on our Manuscript ID: molecules-2200950. We thank you for your insightful comments, which enabled us to improve the manuscript. According to the reviewer’s suggestions, the manuscript has been revised carefully, and the detailed corrections raised by the reviewer have addressed point by point in the revised manuscript as follows.
Reviewer 1
This manuscript reports anti-fungal efficacy in experimental oral candidiasis in immunosuppressed rats in relation to chemical profile. Therefore the identification of the constituents is very important.
The constituents have been identified based on only LC-QTOF-MS/MS data without comparing with data of authentic samples. Without direct comparison with authentic samples, the location of sugar unit can not be determined.
Thus, this manuscript does not contain sufficient data, which means that "Major revision" or "Reject" is adequate for this manuscript.
Response: Many thanks for this observation. The locations and types of sugars were identified as we compared the Rt values, mass spectrum and chromatography with an authentic standard the main compounds such as Kaempferol-3-O-(6-p-coumaroyl)-glucoside, Kaempferol-3,7-O-bis-α-L-rhamnoside, Kaempferol-3-O-glucoside, Luteolin-8-C-glucoside, Kaempferol-3-O- α -L-rhamnoside (Afzelin), Maritimetin-6-O-glucoside, Kaempferol-7- O-neohesperidoside, kaempferol-3-O-robinoside-7-O-rhamnoside, and Kaempferol-3-O- di-glucoside under the same LC-MS conditions. Also, according to Zhuan-Hong, Li., et al [22]. Statement on page 10, lines 277-288.
Reviewer 2 Report (New Reviewer)
This study demonstrated that this study was carried out to study the anti-candidal activity of C. mulare fresh juice and its relation to the immune system using the oral candidiasis model. As well as studying the chemical profile of the juice using by HPLC/QTOF-HR-MS/MS analysis. The study is well designed and data is interesting. However, the manuscript presented some lack of discussion and unclear expression, as a result, overall presentation seems to be less attractive. There are few grammar mistakes and writing/wording issues. The whole work is complete, scientific and practical, and it can be published in this journal after minor revision.
Abstract:
1- The sentence "Profiling of the 2ry metabolites of Chenopodium murale revealed the existence of 42 phytoconstituents including 6 phenolic acids, 26 flavonoids, 1 anthocyanin, 1 Aurone O-glycosides, 1 dihydrochalcone, 5 fatty acids, 3 sugars and 2 organic compounds". Should be correct.
2-The sentence "Meanwhile, juice at 1000 mg/kg for seven days leads to a more significant reduction in mean CFU/ swab". Should be "Meanwhile, juice at 1000 mg/kg for seven days was leads to a more significant reduction in mean CFU/ swab".
3- The discussion is long and needs to be shortened and not expanded unnecessarily.
Author Response
Author's Response
Responses to reviewers’ comments for “Chenopodiastrum murale juice shows anti-fungal efficacy in experimental oral candidiasis in immunosuppressed rats” (Manuscript ID: molecules-2281968)
Dear Reviewer 2,
We are very pleased to receive your letter containing the reviewers’ comments on our Manuscript ID: molecules-2200950. We thank you for your insightful comments, which enabled us to improve the manuscript. According to the reviewer’s suggestions, the manuscript has been revised carefully, and the detailed corrections raised by the reviewer have addressed point by point in the revised manuscript as follows.
Reviewer 2
This study demonstrated that this study was carried out to study the anti-candidal activity of C. mulare fresh juice and its relation to the immune system using the oral candidiasis model. As well as studying the chemical profile of the juice using by HPLC/QTOF-HR-MS/MS analysis. The study is well designed and data is interesting. However, the manuscript presented some lack of discussion and unclear expression, as a result, overall presentation seems to be less attractive. There are few grammar mistakes and writing/wording issues. The whole work is complete, scientific and practical, and it can be published in this journal after minor revision.
Abstract:
Comment 1: The sentence "Profiling of the 2ry metabolites of Chenopodium murale revealed the existence of 42 phytoconstituents including 6 phenolic acids, 26 flavonoids, 1 anthocyanin, 1 Aurone O-glycosides, 1 dihydrochalcone, 5 fatty acids, 3 sugars and 2 organic compounds". Should be correct.
Response: The sentence has been corrected (please see page 1, line 32)
Comment 2: The sentence "Meanwhile, juice at 1000 mg/kg for seven days leads to a more significant reduction in mean CFU/ swab". Should be "Meanwhile, juice at 1000 mg/kg for seven days was leads to a more significant reduction in mean CFU/ swab".
Response: The statement was corrected.
Comment 3: The discussion is long and needs to be shortened and not expanded unnecessarily.
Response: The discussion was reduced from 2475 words to 1897 words.
Reviewer 3 Report (New Reviewer)
The reviewed manuscript is dealing with the application of juice from Chenopodiastrum murale leaves on experimental oral candidiasis in immunosuppressed rats.
Although the research was well planned and executed it was poorly presented.
First of all, the authors should choose which plant name Chenopodiastrum murale or Chenopodium murale they want to use throughout the text and in the abstract.
Use proper abbreviations. For example, one could use SM as an abbreviation for secondary metabolites.
The references should be relevant to the study. For example, reference 22 was not appropriate.
The figures should be corrected and optimized. For example, in figure 1 there are missing letters A, B, and C. The text right after the figure should go in the figure caption.
The MS/MS identification should be checked. How were the sugars identified? What was evidenced in MS/MS spectra that the hexose is glucose but not galactose or mannose? This was also valid for pentoses. How did the authors locate the position of acyl moiety in the molecule? These needed proper discussion.
How did the juice from Chenopodiastrum murale leaves was applied? It was not clear.
Author Response
Author's Response
Responses to reviewers’ comments for “Chenopodiastrum murale juice shows anti-fungal efficacy in experimental oral candidiasis in immunosuppressed rats” (Manuscript ID: molecules-2281968)
Dear Reviewer 3,
We are very pleased to receive your letter containing the reviewers’ comments on our Manuscript ID: molecules-2200950. We thank you for your insightful comments, which enabled us to improve the manuscript. According to the reviewer’s suggestions, the manuscript has been revised carefully, and the detailed corrections raised by the reviewer have addressed point by point in the revised manuscript as follows.
Reviewer 3
The reviewed manuscript is dealing with the application of juice from Chenopodiastrum murale leaves on experimental oral candidiasis in immunosuppressed rats. Although the research was well planned and executed it was poorly presented.
Comment 1: First of all, the authors should choose which plant name Chenopodiastrum murale or Chenopodium murale they want to use throughout the text and in the abstract.
Response: We thank the reviewer for this observation. We used the word Chenopodium murale all over the manuscript and deleted Chenopodiastrum murale.
Comment 2: Use proper abbreviations. For example, one could use SM as an abbreviation for secondary metabolites.
Response: We thank the reviewer for this observation. We use proper abbreviations in the manuscript.
Comment 3: The references should be relevant to the study. For example, reference 22 was not appropriate.
Response: Thanks again for this observation. The references have been checked. Reference 22 has been replaced by another relevant to the study.
Comment 4: The figures should be corrected and optimized. For example, in figure 1 there are missing letters A, B, and C. The text right after the figure should go in the figure caption.
Response: Thanks again for this observation. The missing letters were put on the figures.
Comment 5: The MS/MS identification should be checked. How were the sugars identified? What was evidenced in MS/MS spectra that the hexose is glucose but not galactose or mannose? This was also valid for pentoses. How did the authors locate the position of acyl moiety in the molecule? These needed proper discussion.
Response: The locations and types of sugars were identified as we compared the Rt values, mass spectrum and chromatography with an authentic standard the main compounds such as Kaempferol-3-O-(6-p-coumaroyl)-glucoside, Kaempferol-3,7-O-bis-α-L-rhamnoside, Kaempferol-3-O-glucoside, Luteolin-8-C-glucoside, Kaempferol-3-O- α -L-rhamnoside (Afzelin), Maritimetin-6-O-glucoside, Kaempferol-7- O-neohesperidoside, kaempferol-3-O-robinoside-7-O-rhamnoside, and Kaempferol-3-O- di-glucoside under the same LC-MS conditions. Also according to Zhuan-Hong, Li., et al [22]. The needed proper discussion has been written in detail and compared with other relevant studies.
Comment 6: How did the juice from Chenopodiastrum murale leaves was applied? It was not clear.
Response: Chenopodium murale juice was typically applied on tongue and mouth parts using dropper 1ml like that of Nystatin. The answer on page 23, lines 606-607.
Reviewer 4 Report (New Reviewer)
I found the manuscript " Chenopodiastrum murale juice shows anti-fungal efficacy in experimental oral candidiasis in immunosuppressed rats in Relation to its Chemical Profile" by -Newary et al. interesting, which Chenopodium fought Candida through four strategies, including i) promotion of classical phagocytosis of neutrophils, ii) activation of T cells that activate IFN-γ, IL-2, and IL-17, iii) increasing production of cytotoxic NO and H2O2 that can kill Candida, iv) activation SOD that converts superoxide to antimicrobial materials. These activities
could be due to its active constituents that are documented as an anti-fungal.. The topic is of current interest and suited for the journal; anyways, some modifications of the submitted paper are recommended before publication.
Author Response
Author's Response
Responses to reviewers’ comments for “Chenopodiastrum murale juice shows anti-fungal efficacy in experimental oral candidiasis in immunosuppressed rats” (Manuscript ID: molecules-2281968)
Dear Reviewer 4,
We are very pleased to receive your letter containing the reviewers’ comments on our Manuscript ID: molecules-2200950. We thank you for your insightful comments, which enabled us to improve the manuscript. According to the reviewer’s suggestions, the manuscript has been revised carefully, and the detailed corrections raised by the reviewer have addressed point by point in the revised manuscript as follows.
Reviewer 4:
I found the manuscript " Chenopodiastrum murale juice shows anti-fungal efficacy in experimental oral candidiasis in immunosuppressed rats in Relation to its Chemical Profile" by -Newary et al. interesting, which Chenopodium fought Candida through four strategies, including i) promotion of classical phagocytosis of neutrophils, ii) activation of T cells that activate IFN-γ, IL-2, and IL-17, iii) increasing production of cytotoxic NO and H2O2 that can kill Candida, iv) activation SOD that converts superoxide to antimicrobial materials. These activities could be due to its active constituents that are documented as an anti-fungal.. The topic is of current interest and suited for the journal; anyways, some modifications of the submitted paper are recommended before publication.
Response: We thank reviewer for helping us to improve our manuscript and we welcome any correction.

Round 2
Reviewer 1 Report (Previous Reviewer 1)
I think the problems pointed out previously have been revised.
Author Response
Thanks to Reviewer 1
Manuscript title “Chenopodiastrum murale juice shows anti-fungal efficacy in experimental oral candidiasis in immunosuppressed rats” (Manuscript ID: molecules-2281968)
Dear Reviewer 1,
Thank you for your kind cooperation and helpful assistance, which improved the work and raised its scientific level while also providing the authors with a scientific addition. We will add this information in the next publication.
Finally, we are very grateful to you.
Reviewer 3 Report (New Reviewer)
Definitely, the quality of the manuscript was improved compared to the earlier version. It still requires some minor polishing. The authors use frequently "Chenopodium" as a substitute for Chenopidium murale juice. This should be avoided because it means genus Chenopodium. Instead, they should have an abbreviation for Chenopidium murale juice. For example CMJ or something like that.
Author Response
Author's Response
Responses to reviewers’ comments for “Chenopodiastrum murale juice shows anti-fungal efficacy in experimental oral candidiasis in immunosuppressed rats” (Manuscript ID: molecules-2281968)
Dear Reviewer 3,
We are very pleased to receive your letter containing the reviewers’ comments on our Manuscript ID: molecules-2200950. We thank you for your insightful comments, which enabled us to improve the manuscript. According to the reviewer’s suggestions, the manuscript has been revised carefully, and the detailed corrections raised by the reviewer have addressed point by point in the revised manuscript as follows.
Comment: Definitely, the quality of the manuscript was improved compared to the earlier version. It still requires some minor polishing. The authors use frequently "Chenopodium" as a substitute for Chenopodium murale juice. This should be avoided because it means genus Chenopodium. Instead, they should have an abbreviation for Chenopodium murale juice. For example CMJ or something like that.
Response: Thank you for your kind cooperation and helpful assistance, which improved the work. The required comment was made throughout the manuscript (with yellow shade). The scientific name of Chenopodium murale juice was placed in title of the tables and figures with the abbreviation; CMJ, to facilitate their separate reading without referring to the manuscript.
This manuscript is a resubmission of an earlier submission. The following is a list of the peer review reports and author responses from that submission.
Round 1
Reviewer 1 Report
This manuscript reports biological activity of Chenopodiastrum mulare juice. The biological activity is not chemical information, which means this manuscript is not suitable to Molecules, a journal of chemistry.
Of course, the information of the constituents is chemical and is suitable for Molecules. In this manuscript, however, the constituents have been identified based on only fragment information in LC-MS spectrum, without isolating any constituent nor comparing the data with a known authentic sample. The fragment information in MS spectrum can not identify the absolute configuration of the sugar moiety nor the position of the sugars. Thus, identification based on comparing only LC-MS data is not enough chemical information.
In addition, the fragment data used for identification of constituents seems to contain large ambiguity. The many fragment ions in Figure 3 seem to be baseline noise.
The activity of isolated constituents contains chemical information because it correlates the activity and the chemical structure. But this manuscript does not contain the activity of any isolated compound, which means this manuscript not to contain chemical information.
From the above reasons I would like to suggest the authors to identify the constituents based on enough data and resubmit the revised manuscript to Molecules, or to revise and submit any other journal informing biological activity.